# Study on Factors of People’s Wearing Masks Based on Two Online Surveys: Cross-Sectional Evidence from China

**DOI:** 10.3390/ijerph18073447

**Published:** 2021-03-26

**Authors:** Ling Zhang, Sirong Zhu, Hui Yao, Mengying Li, Guanglin Si, Xiaodong Tan

**Affiliations:** Department of Occupational and Environmental Health, School of Health Sciences, Wuhan University, Wuhan 430071, China; zhangling0@whu.edu.cn (L.Z.); zsr52592@126.com (S.Z.); 15779681295@163.com (H.Y.); limengying@whu.edu.cn (M.L.); sigl1994@163.com (G.S.)

**Keywords:** wearing masks, psychological changes, quarantine, self-protection

## Abstract

(1) Background: To analyze factors of people’s wearing masks based on two online surveys, and to explore whether living area factor or quarantine status could have an impact on mask-wearing. (2) Methods: Chi-square test and logistic regression analysis were used to explore the influence of different factors on people’s behavior of mask-wearing in the present study. R3.6.2 was used for data cleaning, SPSS 26.0 was used to conduct corresponding statistical analysis, and a two-sided *p*-value less than 0.05 was considered to be statistically significant. (3) Results: According to our study, the proportion of participants who wore face masks was higher than 90.0% in both surveys. Single factor analysis showed that the proportion of people wearing face masks raised with the increase of people’s education, age, and monthly income (Yuan) in both surveys. People who lived in rural areas were less likely to wear masks. Mask-wearing rate was lower in the isolated individuals than in the non-isolated ones. (4) Conclusions: Masks-wearing is one of the effective measures for COVID-19 pandemic prevention and control. After the Chinese government made wearing face masks mandatory in public places, most residents in China had developed the habit of wearing masks, contributing to the high rate of masks-wearing in China. However, people in rural areas need to raise their awareness of wearing masks. Meanwhile, the mask-wearing of the isolated individuals needs to be enhanced.

## 1. Introduction

The Coronavirus Disease (COVID-19) pandemic caused by Severe Acute Respiratory Syndrome Coronavirus 2 (SARS-CoV-2) has brought a severe threat to human health. Mainly spread through droplets, close contact, and aerosols, masking, handwashing, and wearing gloves are proved to be able to reduce and prevent the transmission [1]. Most countries and cities made mask-wearing mandatory in public places to limit the transmission. On the one hand, the use of masks can protect people from transmission by trapping larger droplets [2,3], preventing the inhalation of respiratory pathogens and reducing the hand-to-face contact [4]. On the other hand, mask-wearing when sick may reduce the transmission of virus to protect others [5].

Vincent’s study [6] shows that wearing a mask may contribute to the control of COVID-19 by reducing the amount of emission of infected saliva and respiratory droplets from individuals with subclinical or mild COVID-19. In countries with cultural norms or government policies supporting public mask-wearing, per-capita coronavirus mortality increased on average by just 16.2% each week, as compared with 61.9% each week in the remaining countries [7]. For example, many young people are more liberal, and they consider wearing a mask to be a restriction on their freedom and not in line with traditional attitudes [8,9]. Wearing masks is meaningful, but there are still some people who do not wear masks on their daily trips. Besides government policies and regulations, there are many factors that influence people to wear masks. Previous studies indicated that gender could shape mask wearing adherence. For example, with respect to the COVID-19 pandemic, Duarte R and Furtado I [10] found that men were more likely to view mask wearing as “shameful” than women. In another study, the possibility of an individual wearing a mask increased significantly with age, and females are 1.5 times more likely to wear a mask than males. Additionally, the odds of observing a mask on an urban or suburban shopper were similar to 4 times that for rural areas [11]. During different phases of the COVID-19 epidemic, countries widely adopted isolation policies such as self-organized citizen communities [12] and mass quarantine [13], which also had an impact on the behavior of people wearing masks [14]. However, there were few studies on the effect of quarantine on mask wearing rates. In fact, Chinese concentration of quarantine crowds [1] is that (1) confirmed patients; (2) suspected patients; (3) febrile patients; and (4) close contacts are required to wear masks during isolation. These people who are at high risk of transmitting COVID-19 need to wear masks more frequently.

On 22 January 2020, experts from the National Health Commission of China pointed out that Coronavirus Disease could be transmitted from person-to-person; thus, the “mask order” became prevalent in the mainland of China. We conducted this study on the factors of people wearing masks, and in particular, the influence of isolation and regional factors on the wearing masks. Our survey was carried out in the period just after COVID-19 outbreaks and the two surveys were conducted during the quarantine period and just after the resumption work period. This is not available in other cross-sectional studies. We focus on the factors that influence people to wear masks in the immediate aftermath of an outbreak and provide theoretical and policy guidance for countries around the world to slow down the outbreak.

## 2. Materials and Methods

### 2.1. Data Resource

Using self-designed questionnaires, we conducted two surveys to investigate the residents’ psychological state and daily life during the epidemic. The first survey was conducted from 1 February to 5 February, the second week after Wuhan sealed off the city. The second survey was conducted from 20 February to 25 February, the first week after workers returned to work. Based on previous research and compiled after the discussion of experts, the questionnaires were mainly divided into three parts: (1) changes of psychological status, (2) changes of daily life, and (3) demographic characteristics. We conducted the investigations in the form of an online questionnaire of the most widely used application in mainland China, which can quickly reflect the residents’ status and achieve the purpose of this study. We conducted whole-group sampling in three separate areas: Wuhan, Hubei; other cities in Hubei; and other provinces and cities. Invalid and incomplete questionnaires were excluded.

This study mainly used the data of people’s protective measures in daily life part and their sociodemographic information. A total of 3284 and 4071 questionnaires were collected, covering all provinces in mainland China. Our study mainly focused on the ordinary non-segregated population in China. To ensure the suitability of this study, respondents who were medical staff were excluded. Exclusion criteria included: (1) questionnaires that were not logical before or after the survey; (2) questionnaires completed by young people under the age of 18 without assurance that they were in an independent situation; (3) medical staff; and (4) questionnaires from Hong Kong, Macau, and Taiwan were excluded, and only provinces in mainland China were included. Finally, 3104 and 3657 valid questionnaires were obtained. This study was approved by the research ethics committees of Wuhan University (Approval No. 2020YF0064). All participants provided informed consent.

### 2.2. Methods and Design

Chi-square test and logistic regression analysis were used to explore the influence of different factors on people’s behavior of mask-wearing in the present study. Based on the reviewed relevant literature, we included and excluded variables. A preliminary statistical description of the variables was shown (Table 1). R3.6.2 was used for data cleaning, SPSS 20.0 was used to conduct the corresponding statistical analysis, and Stata SE16.0 was used to make graphic, and a two-sided *p*-value less than 0.05 was considered to be statistically significant.

## 3. Results

### 3.1. Demographic Characteristics of Samples

The sociodemographic characteristics of the participants in two surveys were shown in Table 1. Among all participants in the two surveys, most participants are in the 30–39 age group (30.2%), and 2.1% of the participants are over 60 years old. A total of 2218 (32.8%) participants are male and 4543 (67.2%) are female. Of the participants, 96.2% have accepted the education of senior high school or above. The majority (68.8%) are married, and only 261 (3.9%) are living alone. Almost one-third of the respondents earned 2000 to 5000 Yuan per month. Most participants (81.1%) have a job. More than two-thirds of the respondents were in urban areas. Participants who were under non-quarantine account for the vast majority (73.0%) (Table 1).

### 3.2. Behaviors of Masks-Wearing in Different Population

Through the results of the survey, the rate of behaviors of wearing face masks in China is at a relatively high level. As it is shown in Table 2, the difference between of gender and wearing mask was not statistically significant. About age, a higher proportion of older people over 60 years of age would wear masks, and the differences in mask wearing rates between age groups were statistically significant. Respondents who were not in isolation and who are living in urban areas were more likely to wear a mask. Higher income and married groups were more likely to wear masks and the differences were statistically significant across income groups. In all surveys, gender, place of residence, whether one lives alone or not, and the presence of infected people in the vicinity did not affect the rate of mask use (Table 2).

As it was shown in Figure 1, we compared the age distribution of respondents in two surveys. In both surveys, there was an upward trend in mask-wearing rates with age, but the difference was that the trend in wave 1 shows a small peak (97.6%). That means the 40–49 age group was higher than in the 50–59 age group. The wave 2 showed a significantly higher rate of mask wearing than the wave 1, with the exception of the >60 years age group (100% > 97.9%) (Figure 1).

As it was shown in Figure 2, people’s rate of mask wearing all degrees was higher than 90%. In wave 1, the rate of mask wearing in middle school was lower than in high school (1%), and the rate of mask wearing in wave 1 was significantly lower than in wave 2. There has been a steady trend in the rate of wave 2 (Figure 2).

As it was shown in Figure 3, the proportion of different population mask-wearing increases with the increase of their monthly income. Besides, the proportion of participants wearing face masks in the second survey was about 1% higher than that of the first survey, no matter how much they own (Figure 3).

### 3.3. Results of Logistic Regression Analysis

As it was shown in Table 3, six models were constructed to study the influencing factors of people’s wearing masks. Models 1–3 were analyzed from wave 1, and models 4–6 were analyzed from wave 2. Model 1 and model 4 are variables that included basic demographic characteristics, models 2 and model 5 included are variables based on basic demographic characteristics, and models 3 and model 6 add the quarantine variable into the model (Table 3).

In the first survey, we included demographic characteristics variables, where differences in gender, education, and marital status made a difference in mask-wearing (*p* < 0.05). Women are 1.544 times (95%CI (1.061, 2.247)) more likely to wear a mask than men; high school, college, and master’s degree and above were 3.0804 times (95%CI (1.661, 8.711)), 2.750 times (95%CI (1.433, 5.276)), and 4.424 times (95%CI (1.912, 10.238)) more likely to wear a mask than education middle school or below, and 1.967 times (95%CI (1.201, 3.224)) more likely to wear a mask for those who were married than un-married.

After adding the area variable, as shown in model 2, the effect of gender on mask wearing is no longer significant, marital status is significant (married is 1.704 times (95%CI (1.017, 2.855)) more likely to wear a mask than unmarried), and the rate of mask wearing in rural areas is 0.262 times (95%CI (0.172, 0.399)) fewer than in urban areas. With the addition of the area variable, there was little effect for the marriage variable on mask wearing rates. After continuing to add the quarantine variable, only the differences between the quarantine variables and area are significant; rural is 0.262 times (95%CI (0.172, 0.399)) less likely to wear a mask than urban, and the isolation people is 0.629 times (95%CI (0.433, 0.913)) less likely to wear a mask than non-isolation. Quarantine status had no effect on mask-wearing rates between rural and urban areas.

Wave 2 was conducted after unblocking. In model 4, only the effect of “monthly income” on mask-wearing was significant in the demographic characteristics section (2.348 times (95%CI (1.085, 5.085)), 3.085 times (95%CI (1.238, 7.686)), and 3.504 times (95%CI (1.063, 11.557)) more than the <2000 group separately); with the addition of the area variable (model 5), the effect of the work variable on mask-wearing rates was significant (2.307 times more with work than without work); for cities in other provinces, mask-wearing rate was 1.842 times higher than that in Wuhan. After continuing to add the quarantine variable (model 6), the results for the “monthly income” variable on the rate of mask wearing were again significant, with a higher rate of mask wearing for the high-income group. The isolation variable had no effect on mask-wearing rate for area variable.

## 4. Discussion

According to our study, the proportion of participants mask-wearing was higher than 90.0% in both surveys, with 95.9% in the first survey and 96.8% in the second survey, respectively. The proportion of the wave 2 was higher than the wave 1, and it can illustrate that with the progress of making face masks mandatory, people’s awareness of self-protection has increased, and wearing face masks has become people’s living habits. It should be noted that wearing face masks was not people’s daily life habit in China before the epidemic [15]. The previous rate of Chinese residents wearing face masks was not high, and even for medical staff, the compliance of mask-wearing had not reached such a high level. According to a national study of Chinese residents’ environmental exposure behavior patterns (N = 91,121) in 159 monitoring points in 31 provinces, autonomous regions, and municipalities from 2011 to 2012, 16.05% Chinese residents have the habits of wearing face masks, with 17.42% in urban areas and 14.89% in rural areas, and 3.89% of males and 12.16% of females [16]. As of medical staff, according to previous studies, the accuracy rate of mask-wearing need to be further improved [17].

In the single factor analysis, the proportion of mask-wearing raised with the increase of people’s education, age, and monthly income (Yuan) in two surveys. Gender, geographical location factors (city of current residence), etc., have no statistical difference. This is inconsistent with the M. C. Howard study [18]. In a multiple logistic regression in the first survey (model 1), women were 1.544 times more likely than men to wear masks. Highly educated ones and married individuals were more likely to wear masks, yet the effect of gender was not statistically significant after the geographical location factors (city of current residence) and administrative region division factors (urban and rural division) were controlled for model 1. The proportion of mask-wearing in rural areas was 0.262 times lower than in urban areas. This may attribute to two reasons. On the one hand, the promotion and publicity of mask-wearing in urban areas were more intense. Individuals in urban areas have higher health literacy and stronger awareness of protection, and generally accept this habit of health. On the other hand, there are more dense populations in the urban area, in which people travel outside more frequently, so they preferred wearing facemasks more. Model 3 shows that quarantine was also a factor on people wearing masks. The isolated ones were 0.629 times less likely to wear a mask than the non-isolated. This indicates that the quarantine factor reduced the use of masks during the first survey; it may also indicate that people who were in isolation tended to accept our survey.

In a multiple logistic regression of the second survey (model 4), the effect of monthly income on mask-wearing rate was statistically different by controlled social economic status factors (education, work status, and monthly income). The higher the monthly income, the higher the rate of wearing a mask. The reason may be due to the fact that high-income people are busier and the population mobility is more frequent. After adding regional factors, people who have jobs were more likely to wear masks. People who are outside Hubei province were more likely to wear masks. People who were living in urban areas were more likely to wear masks, so at the first. However, the quarantine factor is disappeared gradually with people working again. The first survey was conducted during the second week after Wuhan sealed off the city, while the second investigation was conducted during the first week after workers returned to work. During the second investigation, due to the serious epidemic situation in Hubei Province, work had not been resumed yet, and people in Hubei province travel less, so the possibility of wearing face masks was lower than that in other regions.

We hypothesized that quarantine is one of the factors affecting mask wearing previously. Both single factor analysis and multiple logistic regression analyses indicate that the frequency of mask-wearing decreases for people in quarantine; the frequency of mask-wearing was lower in rural areas than in urban areas, suggesting that the degree of regulation of mask-wearing should be enhanced toward rural and those in isolation. This is consistent with the study of T. Callaghan’s and Xuyu Chen’s study [19,20]. For example, health promotion communication sessions on mask-wearing can be conducted in rural areas to enhance their health awareness. What is more, the risk of the epidemic presence in rural areas is presenting [21].

## 5. Conclusions

Mask-wearing has been proven to be economical and affordable to prevent COVID-19. That is why mask-wearing rates are high in China. Though initially enforced by the Chinese government: for example, mask is required for access to transportation and for going eating, our study shows that there are still some problems with mask-wearing rates in China: mask wearing rates for the isolated individuals are lower than non-isolated ones. This problem has been neglected by relevant authorities. The mask-wearing rate in rural areas is lower than that in urban areas, and there is a need to strengthen the promotion of mask-wearing in rural areas due to the risk of hidden epidemics in rural areas [21].

## 6. Theoretical and Practical Implications

When COVID-19 caused a world pandemic, there was a visible contrast between the responses of citizens in east Asia and other countries [22]. Due to cultural differences and other factors, the compliance of residents in different countries to wear face masks may be different. Some research [6,23] has shown that the usage of face mask by the healthy population in the community to reduce risk of transmission of respiratory viruses remains controversial. For example, wearing a mask may give a false sense of security and make people adopt a reduction in compliance with other important infection control measures [24]. What is more, people must avoid touching their masks and adopt other management measures [25]. Currently, mask-wearing is economical and convenient compared to other ways [26]. Theoretical and empirical results have proven the protective effect of face mask wearing and the urgency of universal masking implementation [27]. Governments in other countries may learn from it and encourage people to wear face masks. Despite the good results of prevention and control, people should continue to wear face masks for asymptomatic people also have the risk of disease transmission. Previous study has found that patients may be contagious before they develop symptoms or even before they find themselves infected for maximal viral shedding of the virus occurs early in the course of the illness [28]. Immediate face mask wearing is recommended. There should also be official guidelines for correct use, and awareness that wearing face masks is not only for pure self-protection, but also protecting one’s community.

## Figures and Tables

**Figure 1 ijerph-18-03447-f001:**
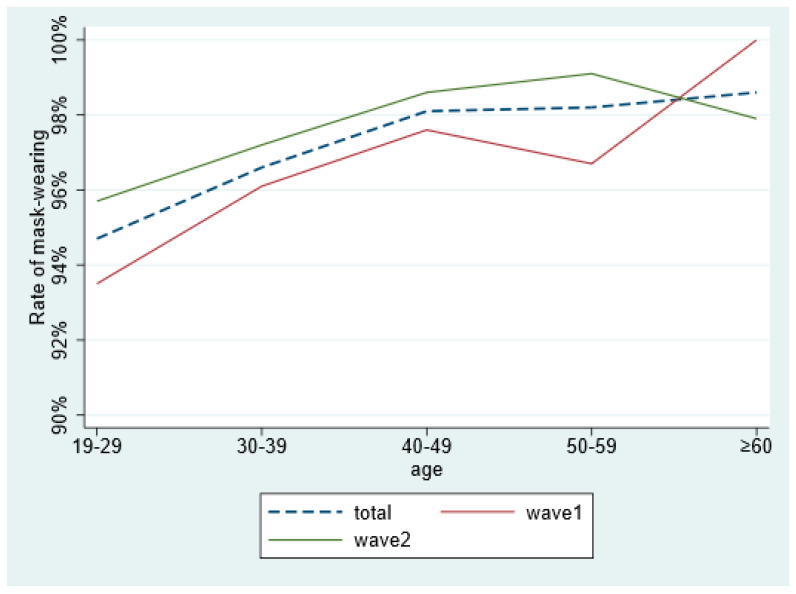
Comparison of the rate of wearing masks in respondents with different age in two surveys.

**Figure 2 ijerph-18-03447-f002:**
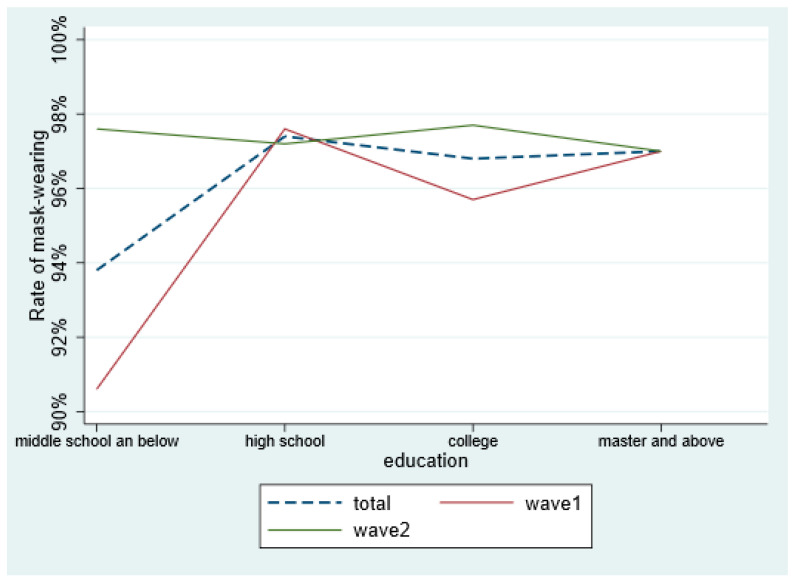
Comparison of the rate of wearing masks people with different education in two surveys.

**Figure 3 ijerph-18-03447-f003:**
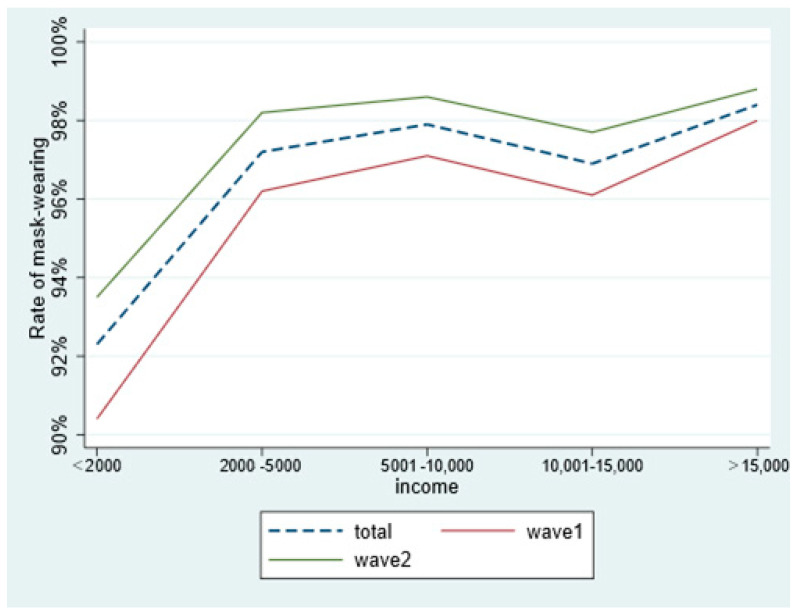
Comparison of the rate of wearing mask people with monthly income in two surveys.

**Table 1 ijerph-18-03447-t001:** Sociodemographic characteristics of participants in the two surveys.

Variable	Variable Assignment/Range	Total (N = 6761)	Survey 1 (*n* = 3104)	Survey 2 (*n* = 3657)	χ2	*p*
Sex					11.169	<0.001
	“Male = 1”	2218 (32.8)	954 (30.7)	1264 (34.6)		
	“Female = 2”	4543 (67.2)	2150 (69.3)	2393 (65.4)		
Age					52.800	<0.001
	19–29	1822 (26.9)	814 (26.2)	1008 (27.6)		
	30–39	2040 (30.2)	1044 (33.6)	996 (27.2)		
	40–49	1823 (27.0)	834 (26.9)	989 (27.0)		
	50–59	936 (13.8)	369 (11.9)	567 (15.5)		
	≥60	140 (2.1)	43 (1.4)	97 (2.7)		
Area					0.027	0.869
	“Urban = 1”	5349 (79.1)	2453 (79.0)	2896 (79.2)		
	“Rural = 2”	1412 (22.3)	651 (21.0)	761 (20.8)		
Current residence					360.001	<0.001
	“Wuhan, Hubei = 1”	1912 (28.3)	619 (12.3)	1293 (35.4)		
	“Other cities in Hubei = 2”	850 (12.6)	267 (5.1)	583 (15.9)		
	“Other provinces and cities = 3”	3999 (59.1)	2218 (82.7)	1781 (48.7)		
Education					54.189	<0.001
	“Middle school or below = 1”	274 (4.1)	149 (5.5)	125 (4.3)		
	“High school = 2”	882 (13)	453 (13.9)	429 (11.8)		
	“College = 3”	4371 (64.7)	2035 (66.4)	2336 (64.9)		
	“Master degree and above = 4”	1234 (18.3)	467 (14.2)	767 (19.1)		
Marital status					0.298	0.585
	“Single = 1”	2112 (31.2)	980 (31.6)	1132 (31.0)		
	“Married = 2”	2404 (68.8)	2124 (68.4)	2525 (69.0)		
Monthly income (Yuan)					31.061	<0.001
	<2000	1042 (15.4)	397 (12.8)	645 (17.6)		
	2000–5000	2171 (32.1)	1034 (33.3)	1137 (31.1)		
	5001–10,000	1967 (29.1)	939 (30.3)	1028 (28.1)		
	10,001–15,000	815 (12.1)	381 (12.3)	434 (11.9)		
	>15,000	766 (11.3)	353 (11.4)	413 (11.3)		
Living alone					9.440	0.002
	“No = 1”	4934 (73)	1343 (98.5)	2364 (96.5)		
	Yes = 1”	1827 (27)	20 (1.5)	86 (3.5)		
Whether there is a job or not					47.418	<0.001
	“No = 1”	1277 (18.9)	537 (17.3)	740 (82.7)		
	“Yes = 1”	5484 (81.1)	2567 (20.2)	2917 (79.8)		
Living alone					80.504	<0.001
No	“No = 1”	6500 (96.1)	3055 (98.4)	3445 (94.2)		
Yes	“Yes = 1”	261 (3.9)	49 (1.6)	212 (5.8)		
Whether there is a job or not					1013.286	<0.001
No	“No = 1”	4934 (73.0)	1686 (54.3)	3248 (88.8)		
Yes	“Yes = 1”	1827 (27.0)	1418 (45.7)	409 (11.2)		
Confirmed infected in personal network					21.771	<0.001
No	“No = 1”	5586 (82.6)	2637 (85.0)	2949 (80.6)		
Yes	“Yes = 1”	1175 (17.4)	467 (15.0)	708 (19.4)		

**Table 2 ijerph-18-03447-t002:** Results of single factor analysis of wearing face masks on two surveys.

		Wave			
Variable	Category	No	Yes	χ2/*t*	*p*
Gender				2.836	0.092
	Male	3.70%	96.30%		
	Female	3.00%	97.00%		
Age				42.761	<0.001
	19–29	5.30%	94.70%		
	30–39	3.40%	96.60%		
	40–49	1.90%	98.10%		
	50–59	1.80%	98.20%		
	≥60	1.40%	98.60%		
Current residence					
	Wuhan, Hubei	2.60%	97.40%	5.061	0.080
	Other cities in Hubei	4.10%	95.90%		
	Other provinces and cities	3.40%	96.60%		
Area				115.572	<0.001
	Urban	2.00%	98.00%		
	Rural	7.70%	92.30%		
Education				11.311	0.023
	Middle school or below	6.60%	93.40%		
	High school	2.60%	97.40%		
	College	3.20%	96.80%		
	Master degree and above	3.00%	97.00%		
Marital status				15.665	<0.001
	Not married	5.20%	94.80%		
	Married	2.30%	97.70%		
Job status				67.836	<0.001
	No	6.90%	93.1%		
	Yes	2.40%	97.6%		
Living alone				0.745	0.388
	No	3.30%	96.70%		
	Yes	2.30%	97.70%		
Monthly income (RMB)				82.700	<0.001
	<2000	7.70%	92.30%		
	2000–5000	2.80%	97.20%		
	5001–10,000	2.10%	97.90%		
	10,001–15,000	3.10%	96.90%		
	>15,000	1.6%	98.4%		
Quarantine				16.362	<0.001
	No	2.70%	97.30%		
	Yes	4.70%	95.30%		
Confirmed infected in personal network				0.275	0.600
	No	3.30%	96.70%		
	Yes	3.00%	97.00%		

**Table 3 ijerph-18-03447-t003:** Multivariable logistics regression of participants’ wearing a mask (*n* = 6761).

		Wave 1 (*n* = 3104)			Wave 2 (*n* = 3657)		
Category		Model 1	Model 2	Model 3	Model 4	Model 5	Model 6
Gender	Female	1.544 *	1.376	1.396	1.239	1.191	1.186
Age	19–29						
	30–39	0.724	0.596	0.604	0.752	0.670	0.660
	40–49	1.217	0.878	0.861	1.528	1.430	1.398
	50–59	1.076	0.668	0.672	2.929	2.508	2.455
	≥60				1.492	1.282	1.219
Education	Middle school or below						
	High school	3.804 *	2.707 *	2.744	0.541	0.547	0.527
	College	2.75 *	1.498	1.447	0.822	0.625	0.589
	Master degree and above	4.424 *	1.923	1.940	0.573	0.41	0.384
Marital status	Married	1.967 *	1.704 *	1.724	0.784	0.879	0.858
Job status	Have a job	1.749	1.733	1.670	2.001	2.307 *	2.306 *
Monthly income (RMB)	<2000						
	2000–5000	1.508	1.15	1.156	2.348 *	2.149	2.178 *
	5001–10,000	1.852	1.32	1.364	3.085 *	2.437	2.518 *
	10,001–15,000	1.181	0.837	0.843	1.914	1.404	1.468
	>15,000	2.395	1.696	1.808	3.504*	2.154	2.216
Current residence	Wuhan, Hubei						
	Other cities in Hubei		0.822	0.814		0.709	0.705
	Other provinces and cities		0.755	0.698		1.842 *	1.840 *
Area	Rural		0.262 *	0.262 *		0.573 *	0.568 *
Quarantine	Yes			0.629 *			0.617

* *p* < 0.05.

## Data Availability

No new data were created or analyzed in this study. Data sharing is not applicable to this article.

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
