# Peer review of "Study on Factors of People’s Wearing Masks Based on Two Online Surveys: Cross-Sectional Evidence from China"

_ijerph, 2021, doi:10.3390/ijerph18073447_

Round 1
Reviewer 1 Report
Dear Authors, Dear Editors,
The title does not match the content of the article. The article does not talk about people's behavior during isolation, but compares who wears - who doesn't wear masks.
Introduction not adequate for the study.
Methodology:
On what websites was the survey posted?
The online survey may be fast but unreliable.
Basically no sample was selected, but who would answer it well.
What is the survey validation?
How surveys were rejected
Why do surveys from abroad? This distorts the results even more if we have different socio-economic conditions in other countries.
If the study was divided into two time periods, both questionnaires should be directed to the same study group. Then the results could be compared. E.g:
(lines 105-106) "the proportion of people wearing face masks was higher in the second survey than in the first" - the authors studied two different groups that cannot be compared with each other; the selection was not representative; So you cannot infer this way.
If 96% of the population wears masks, it is difficult to find differences who do not wear masks among these 4%; I propose to conduct a study on the barriers that 4% of people do not wear masks. It would be more valuable.
Very poor literature.
The article does not contribute anything interesting for science, it does not deal with the behavior of people during isolation, does not correspond to the section "Health Behavior, Chronic Disease and Health Promotion"
Author Response
Dear Ms. Emerald Liang and Reviewer:
Thank you for your letter and for the reviewers’ comments concerning our manuscript entitled “Study on the Behavior of People’s Wearing Masks during Isolation: Cross-sectional Evidence from China” (ID: ijerph-1118989). Those comments are all valuable and very helpful for revising and improving our paper, as well as the important guiding significance to our researches. We have studied comments carefully and have made correction which we hope meet with approval. Revised portion are marked in red in the paper. The main corrections in the paper and the responds to the reviewer’s comments are as flowing:
Responds to the reviewer’s comments:
1. Response to comment: The title does not match the content of the article. The article does not talk about people's behavior during isolation, but compares who wears - who doesn't wear masks.
Response: We are very sorry for our wrong writing,we have made correction according to the Reviewer’s comments. After analyzing the data and organizing the variables, we felt that factors of People's Wearing Masks' was more appropriate for our study. In particular, the effect on area and isolation variables on the wearing of masks.
- Response to comment: Introduction not adequate for the study.
Response: We are very sorry for our negligence of time span for writing and submitting articles. Because of COVID-19 prevalent in China when we designed and written this letter, the study about wearing mask were fewer. Considering the reviewer’s suggestion, we have referred to recently published literature. Introduction was revised according to the purpose of the study, the recipients of benefits, innovation points and main contributions. We provide a theoretical basis for this research using the latest literatures and the hypothesis was formulated. and I hope that my answers will satisfy the reviewers( please referring introduction).
- Response to comment: On what websites was the survey posted?
Response: Website of the survey published on a widely used application in China, Which were used by all people. We have re-written this part in the submission(line 80).
- Response to comment: The online survey may be fast but unreliable.
Response: It is really true as reviewer suggested that online survey may be fast but unreliable,we took this limitation into consideration at the design period. However, it was the objective constraints during the outbreak that led to only online investigations being carried out at the during quarantine. Next, data from online surveys are not necessarily unreliable, as respondents respond independently during isolation and there is no cross-fertilisation of survey results. However there will be some selection bias, the survey group will be more restricted by the characteristics of the online users and the group represented may be limited.
- Response to comment: Basically no sample was selected, but who would answer it well.
Response:The reviewers' comments were very meaningful and exposed our shortcomings in designing the questionnaire and writing the paper. The current survey was conceivably difficult to carry out during the isolation period, so we did carry out an online survey format. There is a selection bias in sampling by publish the questionnaire in the interpersonal circle, but there are many factors common to people in the same household during isolation, such as checking the progress of the epidemic on their iphones. So we did consider the possibility of such questions, But we don't have the ability to perfect it. and I hope my answers will satisfy the reviewers.
- Response to comment: What is the survey validation?
Response: Thanks for your comments,we have add the approval code into the article(Please referring table 1).
7.Response to comment: How surveys were rejected?
Response: We are very sorry for our writing,we have made correction according to the Reviewer’s comments. Exclusion criteria included: 1) questionnaires that were not logical before or after the survey; 2) questionnaires completed by young people under the age of 18 without assurance that they were in an independent situation; 3)medical staff; 4)questionnaires from Hong Kong, Macau and Taiwan were excluded, and only provinces in mainland China were included.
8.Why do surveys from abroad? This distorts the results even more if we have different socio-economic conditions in other countries.
Response: Thanks for your comments, we think your recommendations are meaningful.
We considered the possibility that the web is an open world and that the questionnaire is widely circulated, with the possibility of dissemination abroad, but we eliminated data from outside China during the data processing stage to ensure that the sample size is representative.
- Response to comment: Very poor literature.
Response: We are very sorry for our wrong writing,we have made correction according to the Reviewer’s comments. We have also updated the references to the latest and influential literature (Please referring references).
- Response to comment: The article does not contribute anything interesting for science, it does not deal with the behavior of people during isolation, does not correspond to the section "Health Behavior, Chronic Disease and Health Promotion".
Response: We are very sorry for our writing, we know that the innovation and significance of the article is not clearly indicated. After adopting guidance from the editors and reviewers, we clarified the significance, innovation and main contributions of the investigation into introduction. Regarding the behavioral factors that were not involved during the outbreak, we feel that wearing a mask is one of the behaviors, as well as a preventive measure. At the same time, study of the factors of the wearing masks is also a tool of responding to the policy under health promotion, which is the use of administrative means to coordinate the various sectors of society and fulfill their responsibilities for health. Wearing a mask is one of the health responsibilities that the Chinese government has strongly urged to be coordinated at all levels (including families and communities) to slow the spread of COVID-19.
We tried our best to improve the manuscript and made some changes in the manuscript. These changes will not influence the content and framework of the paper. And here we did not list the changes but marked in red in revised paper. We appreciate for Editors/Reviewers’ warm work earnestly, and hope that the correction will meet with approval.
Once again, thank you very much for your comments and suggestions.
Reviewer 2 Report
Firstly, I would like to thank the authors for the opportunity of reading and reviewing their manuscript. Although the topic presented in the paper is original and fits with the journal scope, it needs some changes to be adapted to the quality of the IJERPH.
Regarding the manuscript, I have some concerns/suggestions:
- Abstract
From my point of view, the overall quality of the abstract should be improved. Especially, the last paragraph. The authors start with “objective” but I think the content is more related to the conclusions.
- Literature review
I have observed a lack of reference background in this section. I think the authors should choose a theoretical framework or model which can support their results. Also, there is a lack of hypotheses. It is clear that the main objective of the paper is to analyze what factors could predict the use of facemasks, but if the authors could choose a theoretical background the authors could venture their hypotheses. This is highly important, as the manuscript in its current state is lacking a fundamental literature and theoretical background. Otherwise, it is simply a report of the results.
- About materials and methods:
The methodology section needs a better structure. The measures section is missing.
The name of Tables and Figures should start in capital letter.
- Regarding, Results and Discussion Section:
The discussion section is limited to stating whether or not the results are in line with previous studies but does not try to find out why. Again, I defend that if the authors could choose a theoretical model that would help to explain and integrate the main variables, and it would be easier to explain the findings.
A paragraph regarding theoretical and practical implications is needed.
Finally, I think the reference list is poor. As I have mentioned before, there is a lack of theoretical/literature background.
Good luck!
Author Response
Reviewer #2:
Dear Ms. Emerald Liang and Reviewer:
Thank you for your letter and for the reviewers’ comments concerning our manuscript entitled “Paper Title” (ID: ijerph-1118989). Those comments are all valuable and very helpful for revising and improving our paper, as well as the important guiding significance to our researches. We have studied comments carefully and have made correction which we hope meet with approval. Revised portion are marked in red in the paper. The main corrections in the paper and the responds to the reviewer’s comments are as flowing:
Responds to the reviewer’s comments:
- Response to comment: From my point of view, the overall quality of the abstract should be improved. Especially, the last paragraph. The authors start with “objective” but I think the content is more related to the conclusions.
Response: Thanks for your comments, Changes have been made in accordance with your comments.(Line15-29) The modifications are as follows: Results: According to our study, the general proportion of participants wearing face masks in both surveys was higher than 90.0%. Single factor analysis show that the proportion of people wearing face masks raised with the increase of people’s education, age, monthly income (Yuan) in all surveys. People who lived in rural areas will be less likely to wear masks. Mask-wearing rates will be lower in the isolated than in the non-isolated.(4)Conclusions: Masks-wearing is one of tools for epidemic prevention and control, After the mask order issued, Chinese gradually started to develop the habit of wearing masks, so the rate of masks-wearing is high in China. Our study have shown that the awareness of people in rural areas to wear masks still needs to be enhanced; Meanwhile, the mask-wearing of the isolation needs pay attention.
- Response to comment: I have observed a lack of reference background in this section. I think the authors should choose a theoretical framework or model which can support their results. Also, there is a lack of hypotheses. It is clear that the main objective of the paper is to analyze what factors could predict the use of facemasks, but if the authors could choose a theoretical background the authors could venture their hypotheses. This is highly important, as the manuscript in its current state is lacking a fundamental literature and theoretical background. Otherwise, it is simply a report of the results.
Response: Thanks for your suggestions, which we think are practical in our writing. We chose a theoretical framework or model which can support their results, and developed hypotheses based on the background of the literature review. We wondered whether isolation factors and regional factors would have an effect on the frequency of mask wearing. At the same time, We have also updated the references to the latest and influential literature. Wish my response will satisfy you.
- Response to comment: The methodology section needs a better structure. The measures section is missing. The name of Tables and Figures should start in capital letter.
Response: Thanks for your comments. Revisions have been made in accordance with the reviewers' suggestions. methodology section: we have been refined according to a standard methodological template: how the survey was conducted? refined inclusion and exclusion criteria, how the sampling was conducted? The measures section has been added, as shown in Table 1. The name of Tables and Figures were revised in article.
- Response to comment: The discussion section is limited to stating whether or not the results are in line with previous studies but does not try to find out why. Again, I defend that if the authors could choose a theoretical model that would help to explain and integrate the main variables, and it would be easier to explain the findings.
Response: Thank you for your advice, I find it very useful. Changes have been made as you suggested. In the discussion section, we compare the results with previous studies and try to find out the reasons.The approximate process is that there are relatively few previous studies on isolation factors, we presupposed that isolation factors would cause changes in mask wearing rates, and then went through statistical methods to verify that isolation factors in mask wearing rates.
- Response to comment: A paragraph regarding theoretical and practical implications is needed.
Response: Thank you for your suggestions, and changes have been made in accordance with your suggestions. As shown in line264-280. Theoretical implications: it is possible to bring academic attention to the impact of isolation factors on mask-wearing; practical implications:it is for the relevant department to enhance the regulation of mask-wearing in the isolation.
- Response to comment: I think the reference list is poor. As I have mentioned before, there is a lack of theoretical/literature background.
Response: Thanks for your comments. we are very sorry for our negligence of time span for writing and submitting articles. Because of COVID-19 prevalent in China when we designed and written this letter, the study about wearing mask were fewer. Considering the reviewer’s suggestion, we have referred to recently published literature. Introduction was revised according to the purpose of the study, the recipients of benefits, innovation points and main contributions. We provide a theoretical basis for this research using the latest literatures and the hypothesis was formulated. and I hope that my answers will satisfy the reviewers( please referring introduction).
We tried our best to improve the manuscript and made some changes in the manuscript. These changes will not influence the content and framework of the paper. And here we did not list the changes but marked in red in revised paper. We appreciate for Editors/Reviewers’ warm work earnestly, and hope that the correction will meet with approval.
Once again, thank you very much for your comments and suggestions.
Reviewer 3 Report
The article presents an interesting study. The article is interesting and timely but it suffers from various limitations that must be addressed before it is accepted for publication. I recommend a minor revision.
- Introduction. In the introduction, the authors should clearly present the following: What is missing and what is the gap? Why there is a need to conduct this study? Who will benefit? What is the novelty of this work? What are its main contributions? All these questions are unclear to me in the current version of the article.
- The key references and literature related to behavior of people’s wearing masks are missing. Authors have cited too many old social media studies. Please replace those with more recent studies.
- Discussion: Please separate implications from the discussion. The new section should have two sub-sections - Theoretical and practical implications.
Author Response
Dear Ms. Emerald Liang and Reviewers:
Thank you for your letter and for the reviewers’ comments concerning our manuscript entitled “Paper Title” (ID: ijerph-1118989). Those comments are all valuable and very helpful for revising and improving our paper, as well as the important guiding significance to our researches. We have studied comments carefully and have made correction which we hope meet with approval. Revised portion are marked in red in the paper. The main corrections in the paper and the responds to the reviewer’s comments are as flowing:
Responds to the reviewer’s comments:
- Response to comment: Introduction. In the introduction, the authors should clearly sent the following: What is missing andwhat is the gap? Why there is a need to conduct this study? Who will benefit? What is the novelty of this work? What are its main contributions? All these questions are unclear to me in the current version of the article.
Response: Firstly, thank you for guiding me onhow to write background section. By reviewing the influencing factors associated with wearing masks, we found that there is little research on the effect of isolation variables on wearing masks. According to the actual situation in China, there is an interaction between area factors and isolution. These are missing in previous studies. The purpose of the study has been described in more detail within introduction. The beneficiaries will be the humans still affected by the outbreak of COVID-19.Finally,main contributions were explained in the introduction. I hope my answer will satisfy the reviewers.
2.The key references and literature related to behavior of people’s wearing masks are missing. Authors have cited too many old social media studies. Please replace those with more recent studies.
Response: Thanks for your comments, we think your recommendations are meaningful. Based on your suggestion, we referred to the recently published authoritative literature on factors influencing mask wearing rates and removed old social media studies.
3.Discussion: Please separate implications from the discussion. The new section should have two sub-sections - Theoretical and practical implications.
Response: Thanks for your comments. We have divided the discussion into two sub-sections -theoretical and practical implications.
We tried our best to improve the manuscript and made some changes in the manuscript. These changes will not influence the content and framework of the paper. And here we did not list the changes but marked in red in revised paper. We appreciate for Editors/Reviewers’ warm work earnestly, and hope that the correction will meet with approval.
Once again, thank you very much for your comments and suggestions.
Round 2
Reviewer 1 Report
Dear Editors, Dear Authors,
The authors greatly strengthened the positions of the manuscript, it certainly increased its value, it is better written, it constitutes a logical whole.
The article should be checked in terms of language (e.g. typos).
The literature is incomplete - the number of the volume (number), pages, doi, etc. is missing in various places. This should absolutely be corrected before publication.
Certainly, the article is better than the previous one, well… it is a completely new article, I can recommend its publication.
Author Response
Dear Ms. Emerald Liang and Reviewer:
Thank you for your letter and for the reviewers’ comments concerning our manuscript entitled “Study on the Behavior of People’s Wearing Masks during Isolation: Cross-sectional Evidence from China” (ID: ijerph-1118989). Those comments are all valuable and very helpful for revising and improving our paper, as well as the important guiding significance to our researches. We have studied comments carefully and have made correction which we hope meet with approval. Revised portion are marked using “tracker changes”. The main corrections in the paper and the responds to the reviewer’s comments are as flowing:
1.The article should be checked in terms of language (e.g. typos).
Response:Thanks for your comments,we have made changes to language of the article, please see the attached document for details.
2.The literature is incomplete - the number of the volume (number), pages, doi, etc. is missing in various places.
Response:Thanks for your comments,we have made changes to language of the article. Although we acknowledge that partly citation of the literature is still incomplete, we have tried to include a reasonable number of pertinent references.

Reviewer 2 Report
The authors have improved their manuscript according to reviewer's comments
Author Response
Dear Ms. Emerald Liang and Reviewer:
Thank you for your letter and for the reviewers’ comments concerning our manuscript entitled “Study on the Behavior of People’s Wearing Masks during Isolation: Cross-sectional Evidence from China” (ID: ijerph-1118989). Those comments are all valuable and very helpful for revising and improving our paper, as well as the important guiding significance to our researches. We have studied comments carefully and have made correction which we hope meet with approval. Revised portion are marked using "tracker changes". The main corrections in the paper and the responds to the reviewer’s comments are as flowing:
Responds to the reviewer’s comments:
1.The authors have improved their manuscript according to reviewer's comments
Respones:Thanks for your comments,we have changed our manuscript according to reviewer's comments. The revised are listed in the attachment.
